# The Evolutionary History of the *Chymase Locus* -a Locus Encoding Several of the Major Hematopoietic Serine Proteases

**DOI:** 10.3390/ijms222010975

**Published:** 2021-10-11

**Authors:** Srinivas Akula, Zhirong Fu, Sara Wernersson, Lars Hellman

**Affiliations:** 1The Biomedical Center, Department of Cell and Molecular Biology, Uppsala University, Box 596, SE-751 24 Uppsala, Sweden; srinivas.akula@icm.uu.se (S.A.); fuzhirong.zju@gmail.com (Z.F.); 2Department of Anatomy, Physiology and Biochemistry, Swedish University of Agricultural Sciences, Box 7011, SE-750 07 Uppsala, Sweden; Sara.Wernersson@slu.se

**Keywords:** mast cell, tryptase, chymase, serine protease, evolution, granzyme

## Abstract

Several hematopoietic cells of the immune system store large amounts of proteases in cytoplasmic granules. The absolute majority of these proteases belong to the large family of chymotrypsin-related serine proteases. The chymase locus is one of four loci encoding these granule-associated serine proteases in mammals. The chymase locus encodes only four genes in primates, (1) the gene for a mast-cell-specific chymotryptic enzyme, the chymase; (2) a T-cell-expressed asp-ase, granzyme B; (3) a neutrophil-expressed chymotryptic enzyme, cathepsin G; and (4) a T-cell-expressed chymotryptic enzyme named granzyme H. Interestingly, this locus has experienced a number of quite dramatic expansions during mammalian evolution. This is illustrated by the very large number of functional protease genes found in the chymase locus of mice (15 genes) and rats (18 genes). A separate expansion has also occurred in ruminants, where we find a new class of protease genes, the duodenases, which are expressed in the intestinal region. In contrast, the opossum has only two functional genes in this locus, the mast cell (MC) chymase and granzyme B. This low number of genes may be the result of an inversion, which may have hindered unequal crossing over, a mechanism which may have been a major factor in the expansion within the rodent lineage. The chymase locus can be traced back to early tetrapods as genes that cluster with the mammalian genes in phylogenetic trees can be found in frogs, alligators and turtles, but appear to have been lost in birds. We here present the collected data concerning the evolution of this rapidly evolving locus, and how these changes in gene numbers and specificities may have affected the immune functions in the various tetrapod species.

## 1. Background

Cells from several of the hematopoietic cell lineages store large amounts of proteases within cytoplasmic granules [1]. They are stored in an active form within these granules in tight complexes with negatively charged proteoglycans such as heparin and chondroitin sulfate [2,3,4,5]. In a mast cell, these proteases can account for up to 35% of the total cellular protein, and the absolute majority of these belong to the chymotrypsin-related serine protease family [6]. The mRNAs encoding these proteases are also the most highly expressed transcripts in the mature mast cell, where the expression levels can be in the range of several percent of the total transcriptome [7,8,9]. Serine proteases belonging to this family have in mammals been identified in neutrophils, cytotoxic T cells and natural killer (NK) cells, as well as in mast cells and basophils but not in B cells or dendritic cells and only at relatively low levels in eosinophils. A number of very diverse functions have been identified for these proteases, including blood pressure regulation, the induction of apoptosis, the killing of bacteria and fungi, the inactivation of insect and snake toxins, the mobilization or degradation of cytokines and the degradation of connective tissue components [10,11,12,13,14,15,16]. A very broad spectrum of primary cleavage specificities has also been observed, including chymase, tryptase, asp-ase, elastase and met-ase specificities, which highlights the large flexibility in the active site of these proteases [14].

Mast cells (MCs) primarily express enzymes having chymotryptic and tryptic primary cleavage specificities, respectively. These enzymes are therefore named chymases and tryptases. Neutrophils express several enzymes with elastase, tryptase and chymase specificities [14,17,18]. Depending on the species, T cells and NK cells express between five and 17 different granzymes, having tryptase, asp-ase, chymase and met-ase specificities [14,19]. The genes encoding these proteases are organized in four different loci: the *met-ase locus*, the *MC tryptase locus*, the *MC chymase locus* and the *T cell tryptase locus*, also called the *granzyme A/K locus*. These loci are in mammals often located on four different chromosomes, indicating that some of them may originate from whole-genome duplications, so-called tetraploidizations. Two such tertraploidizations most likely occurred during early vertebrate evolution. The original locus was therefore most likely present before these evolutionary important events [20].

In primates and dogs, the MC chymase locus encodes only four functional protease genes: the MC chymase, neutrophil cathepsin G and the T-cell- and NK-cell-expressed granzymes B and H [11,19,21,22]. Interestingly, a massive expansion has occurred within the *chymase locus* in ruminants and rodents. For example, mice have 15 and rats have 18 functional protease genes within this locus [11,19,21,22], and in ruminants (cattle and sheep) there are several members of a new subfamily of enzymes within this locus, the duodenases [19]. Interestingly, although these duodenases originate from genes expressed by hematopoietic cells, they are now expressed primarily in the intestinal region, the duodenum, where they most likely participate in food digestion [23].

In almost all mammals studied except rodents, the *chymase locus* contains only one mast cell expressed chymase gene, which is positioned at one end of the locus, adjacent to the region of non-protease bordering genes (Figure 1). This gene has been designated the α-chymase. A new subfamily of enzymes closely related to the α-chymases has been identified in rodents and these have been designated β-chymases. The β-chymases form a separate sister branch on the phylogenetic tree indicating that they are evolutionary the result of a relatively early gene duplication from the α-chymases. The β-chymases have only been found in rodents with two possible exceptions being cats and dogs (Figure 1).

Rodents, as represented by mice and rats, have a relatively large number of functional β-chymase genes. Mice have five functional β-chymases: mouse mast cell protease (Mcpt) or (mMCP)-1, mMCP-2, mMCP-4, mMCP-9 and mMCP-10 (also named Cma2), whereas rats have seven: rat (r)MCP-1, rMCP-2, rMCP-3, rMCP-4, rMCP-1-like 1 (also named the vascular chymase (Vch)), rMCP-1-like 2 and rMCP-1-like 3 (Figure 1) [19]. These β-chymases are probably the result of an early gene duplication of the α-chymase genes, where this subfamily has then expanded quite extensively through a number of successive duplications both in mice and rats. An additional subfamily of proteases has also appeared within this locus in rodents, the mMCP-8 subfamily [19]. Only one mMCP-8 gene is found in mice, whereas three functional homologues are present in the rat locus: rMCP-8, rMCP-9 and rMCP-10 (Figure 1) [19]. Unlike the α- and β-chymases, which are all expressed by different MC populations, mMCP-8 is exclusively expressed by basophils. In fact, mMCP-8 is the first basophil-specific marker in any species to be cloned [25]. In contrast to those in mice, mMCP-8 related enzymes in rats do not seem to be restricted to basophils as they are also expressed in mucosal MCs [7,26]. No mMCP-8-like gene is present in primates, dogs or ruminants, which is why this subfamily has been suggested to be unique to rodents [19]. However, this notion is now being challenged by recent findings in horses, as described below.

Two major mast cell populations have been identified in both primates and rodents and they have been named after their primary tissue location. One population is found in connective tissues and these mast cells are therefore named connective tissue mast cells (CTMCs). They are primarily found in the skin, in the tongue and in connective tissue around different organs [8]. Interestingly, although mast cells are totally absent in most parts of the brain, CTMCs are present in relatively high numbers in one particular region of the brain, in the fimbria and velum interpositum near the hippocampus, and within the hippocampal formation itself [27]. The second major mast cell population is found in the intestinal mucosa and also in the lungs of primates and these have therefore been named mucosal mast cells (MMCs). These two populations of mast cells display very different protease profiles. CTMCs in mice express the β-chymase mMCP-4, the elastase mMCP-5, the tryptases mMCP-6 and 7 and the mast-cell-specific carboxypeptidase A3 (CPA3), whereas the MMCs only express two chymases, mMCP-1 and 2 [28]. In contrast, in humans MMCs only express tryptase and are therefore named MC_T_, whereas CTMCs express both tryptase and chymase and are named MC_TC_. Notably, MC_TC_ also express cathepsin G and both MC_T_ and MC_TC_ can express CPA3 at the mRNA level, but CPA3 protein is only found in MC_TC_ [29,30].

In addition to the mast cell chymase and cathepsin G, the human chymase locus also harbors two granzyme genes, *grzB* and *grzH*. In rodents, this region of the locus has, similarly to the chymases, experienced a large number of successive gene duplications. These duplications have resulted in seven functional granzyme genes in mice: *GrzB*, *GrzC*, *GrzD*, *GrzE*, *GrzF*, *GrzG* and *GrzN* [19]. The granzymes of the rat locus have undergone an even more extensive set of gene duplications in this region. There are seven different, apparently functional granzyme genes in this region of the rat genome, as well as five pseudogenes. The large numbers of gene duplications observed in the mouse and rat loci have resulted in a major expansion of the entire locus. The rat locus is 15 times larger than the corresponding locus in dogs (Figure 1). The full functional consequences and the evolutionary forces driving this expansion are not yet known. However, the fact that both mice and rats have experienced similar but seemingly independent expansions gives a strong indication that it is a selectable advantage to have a larger number of chymase and granzyme genes in rodents.

The increasing number of sequenced genomes has now opened up the possibility of examining the early events in the evolution of the chymase locus. The more detailed map of this chromosomal region in two frog species, American and the Chinese alligators and a few turtle species have made it possible to trace the origin of this locus to early tetrapods. We here summarize the collected information concerning this rapidly evolving locus and the information concerning the extended cleavage specificity of individual proteases. With this information we can now trace the origin of several of the conserved members of this family of hematopoietic serine proteases. Moreover, we can start analyzing the role of conserved enzyme specificities and hence elucidate the functional consequences of the changes in gene numbers on various immune functions in different tetrapod species.

## 2. Detailed Scale Maps of the *Chymase Locus* in Different Tetrapod Species

To study the appearance and diversification of the *chymase locus* during vertebrate evolution, we have analyzed this chromosomal region in a large panel of different vertebrate species (Figure 1 and Figure 2). By drawing scale maps we can visualize the total size of the locus and the distances between the genes, making it easier to obtain a correct view of the changes that have occurred during evolution.

In all primates analyzed, including human, chimpanzee and Rhesus macaque, we find a small and relatively compact chymase locus with only four genes, all coding for serine proteases. Moreover, the chymase loci of these primates are bordered by the same genes—at one end, close to the mast cell α-chymase, by *RIPK3*, the receptor-interacting serine/threonine-protein kinase 3 gene, which encodes a protein that is part of the TNF receptor-I signaling complex (Figure 1). At this end we also find cerebellin-3 (*CBLN3*), involved in synaptic functions in the brain, *NYNRIN*, a gene containing a retroviral integrase (possibly non-functional) and *SDR39U1*, the gene for a short-chain dehydrogenase (Figure 1). At the other end of the locus, close to the granzyme B gene, we find syntaxin-binding protein 6, *STXBP6* (Figure 1).

In the cat genome we find evidence for the presence of two more genes in the *chymase locus*, in addition to the four genes also found in primates. One of these additional genes (named *MCP1* in Figure 1) is located close to the α-chymase gene and shows sequence similarity to the rodent β-chymases. The cat is so far the only species outside of the rodent branch on the mammalian tree that shows the presence of a β-chymase-like gene. This second chymase gene shows a higher similarity with β-chymases at the nucleotide level, but a higher similarity with the α-chymases at the amino acid level. This gene was also present in an earlier assembly of the dog genome; however, this gene has been removed in the most recent genome update, indicating that the dog genome still requires a detailed reexamination in this region (Figure 1). A second additional gene in the cat locus appears to be a recent gene duplication of granzyme B (named *GzmB-like* in Figure 1).

The artiodactyls, as represented by pigs, sheep and cattle, have experienced a major increase in the number of genes within this locus. The majority of new genes belong to a new subfamily of proteases named duodenases due to their tissue specificity. They are no longer expressed in hematopoietic cells but in the duodenum, where they most likely take part in food digestion [23,31]. There is one such duodenase gene in the pig locus, four in cattle and five in sheep (Figure 1). However, only three out of the four duodenases in the cattle genome seems to be functional. In these three species—cattle, sheep and pigs—we also find duplications of the cathepsin G gene and in cattle and sheep there are also duplications of the α-chymase gene (Figure 1). One or several additional granzyme genes are also found in sheep and cattle but not in pigs (Figure 1). A rearrangement has occurred at one end of the locus in cattle and sheep, but not in pigs, indicating that this rearrangement has occurred after the separation of the pigs from the lineage leading to cattle and sheep (Figure 1). The bordering genes *RIPK3*, *CBLN3*, *NYNRIN* and *SDR39U1* have in cattle and sheep been replaced by another gene region containing the genes *STOML1* and *LOXL1* (Figure 1). *LOXL1*, which is located close to the α-chymase gene, is the gene for Lysyl oxidase like-1, an enzyme that is essential for the first steps in the formation of crosslinks in collagens and elastin. *STOML1* is a gene for stomatin-like protein 1, a protein of yet unknown function.

In the horse genome, we find only a single α-chymase gene and a single cathepsin G gene, and no β-chymase genes. However, a massive expansion of the number of granzyme B and granzyme H genes has occurred. Interestingly, we also find two *mMCP-8-like* genes in the chymase locus of horses that have not been reported earlier (*MCP8-like* and *GzmH-LP2* in Figure 1). This new information changes our previous view of *mMCP-8-like* genes as being rodent-specific [19].

As mentioned earlier, the *chymase locus* of some rodents have experienced a large number of gene duplications, resulting in as many as 15 functional protease genes in the mouse and 18 such genes in the rat. Interestingly, the rat locus has increased in size by 15 times compared to the dog and the entire locus is full of functional genes but also several pseudogenes (Figure 1). In contrast, the *chymase locus* in primates does not contain any detectable traces of pseudogenes. Only one pseudogene is also found in the mouse locus, indicating independent gene amplifications in mice and rats. We can also observe quite extensive increases in gene numbers in golden and Chinese hamsters and in guinea pigs. The situation in the gerbil is not yet possible to delineate as the locus is still very incomplete. The guinea pig chymase locus is also not fully sequenced, but it is most likely almost complete, although the two sequenced parts have not yet been connected (Figure 1). In the rabbit the situation is quite different, as we only find five genes of which one is a pseudogene, *Cma1*. This suggests that the massive gene amplification in rodents did not occur in the early part of the rodent branch of the evolutionary tree (Figure 1). Interestingly also, as seen in the ruminants, one end of the chymase locus in rabbits has experienced a rearrangement (Figure 1). In mice, rats, hamsters and gerbils, the other end of the locus, compared to the ruminants, has experienced a rearrangement, where *STXBP6* has been removed and replaced by the gene *RNF17*, the ring finger protein 17, involved in the regulation of the transcriptional activity of Myc. However, no such rearrangement is observed in guinea pigs and rabbits, indicating that this rearrangement occurred after the separation of the guinea pig ancestor from the other rodents (Figure 1).

A recent update of the rat genome has resulted in quite dramatic changes in the overall structure of the locus. A number of the genes present in the 2014 version, including a large number of pseudogenes, have been removed in the updated 2020 version of the genome, and the locus now contains only six pseudogenes (Figure 1) [21,22]. The large difference between these two versions of the genome sequence is a good example of the difficulties involved in mapping highly repetitive regions of the genome. The reduction in the number of pseudogenes is most likely also a result of the higher quality of the sequence.

When it comes to the non-placental mammals, we can see that the opossum, a marsupial, only has two genes within the chymase locus. The same bordering genes are present as in the primates but one of them—*STXBP6*—has a changed orientation, indicating that an inversion has occurred (Figure 1). This inversion may have been the reason for the lack of expansion, as seen in the rodent lineage. It may have hindered a process of unequal crossing-over that most likely has been the major driving force in the expansion of this locus in rodents.

Several new genome sequences of other marsupials have recently been added to the database. We here include a map of the *chymase locus* of the wombat and the Tasmanian devil (Figure 1). In the wombat, the *chymase locus* is similar to that in the opossum and contains only two genes. However, in the Tasmanian devil the situation is quite different. In contrast to what we observed in the opossum, the separation of the locus into two parts by inversion has in the Tasmanian devil not resulted in a block in gene amplifications. Here, we find a relatively high number of both chymase- and granzyme-related genes (Figure 1). Interestingly, the orientation of the different parts of the locus is different in the Tasmanian devil compared to the opossum and wombat, indicating that there may have been independent rearrangements and that gene duplications may have occurred in the Tasmanian devil before the rearrangement, which have made unequal crossing-over possible within each segment of the locus.

In monotremes, as represented by the platypus, we see the first example of a splitting of the *chymase locus* into two parts, with one part having the same bordering genes on one side as the primate locus, i.e., the side of the α-chymase gene with the bordering gene *SDR39U1* (Figure 1). This part of the locus contains two closely related chymase genes (Figure 1). The second part of the locus contains a granzyme-B-like gene that is surrounded by genes not found in the *chymase locus* of any of the other mammals (Figure 1). The exact position of these two pieces relative to each other is not yet known, as the platypus genome sequence is still incomplete.

When we look at the *chymase locus* in reptiles, birds and amphibians, the situation becomes slightly more complex. Both the American and the Chinese alligator have a classical *chymase locus*, in which at least the American alligator has one of the bordering genes of the human locus, the *SDR39U1*. Both of these alligator species also have another locus of related genes, with *GNA11* and *NCLN* as the bordering genes (Figure 2). In a phylogenetic tree, the genes present in this additional locus form a separate branch and do not cluster with the classical *chymase locus* genes [19]. Interestingly, this locus is not found in any mammal, and was most likely lost when the mammals branched off during early tetrapod evolution. However, this locus was found in all the analyzed birds, alligators, turtles and frogs, indicating that it is an early locus (Figure 2). This locus contains two serine protease genes in the chicken, *CTSG-like* and *GzmG-like*; two in the Chinese alligator, *GzmM* and *GzmE*; and one in the Western clawed frog, *GzmH* (Figure 2).

The classical *chymase locus* in American and the Chinese alligators differs markedly in the number of serine proteases encoded in this locus. The American alligator sequence, which appears to be the most complete, contains only three serine protease genes in the classical chymase locus (Figure 2). In contrast, the Chinese alligator has a total of nine chymase-locus-related serine protease genes, dispersed on three different sequence contigs, indicating a partial sequence (Figure 2). The turtle has a classical chymase locus with one of the bordering genes, *RIPK3*, and 20 different serine protease genes, of which at least five seem to be pseudogenes (Figure 2). In the two amphibians analyzed, the American clawed frog and the Western clawed frog, we can find both the classical *chymase locus* and the related locus found in reptiles and birds (Figure 2). Interestingly, in these two amphibians, these two loci are located within the same chromosomal region relatively close to each other, indicating that they may have been part of the same original locus (Figure 2). In the American clawed frog, we find two protease genes within the classical *chymase locus* and in the Western clawed frog there is only one such gene (marked in light-dark blue in Figure 2).

None of the genes that cluster with the classical *chymase locus* genes in mammals, reptiles and amphibians were found in any of the fish species we have analyzed [19].

## 3. Extended Cleavage Specificities of Chymase-Locus-Expressed Serine Proteases

In an attempt to understand their functions we have analyzed a large number of the chymase-locus-encoded proteases for their extended cleavage specificities. The majority of them have been analyzed via a phage display using a phage T7 library. All the genes we have analyzed so far by phage display are marked with a red star in Figure 1. The library we have used contains approximately 50 million individual clones. The individual clones in this library express nine different amino acid long random regions (nonamers), followed by a six-residue histidine tag fused to a capsid protein of the phage. The His-tag makes it possible to attach the phage to the surface of Ni^2+^-chelating agarose beads and thereby immobilizing the phage for analysis of the selectivity of the different proteases. The protease can then selectively cleave sequences within the nonamer region that it prefers. By means of such a cleavage, the phages are released from the Ni^2+^ beads and can be recovered from the supernatant. By repeating this selection process 5–7 times we obtain a strong selection for those sequences that the protease prefers. By sequencing a panel of individual clones from the last selection round and comparing the nine amino acids in the random region we can obtain information on the extended cleavage specificity of the protease. A schematic drawing of the procedure is shown in an earlier publication [14].

For some of these proteases we have, for unknown reasons, not succeeded in obtaining a good amplification during the panning in the phage display. For a few of these proteases we have then used an alternative strategy to obtain information concerning their extended specificity. A new type of recombinant substrate has been developed in our lab to verify the results from the phage display. By also using such substrates for previously uncharacterized proteases it is possible to obtain quantitative information concerning the role of individual amino acids at various positions at and around the cleavage site. These recombinant substrates are based on the *Escherichia coli* redox protein, thioredoxin (Trx). In the linker region between two Trx molecules we have inserted a short flexible kinker region consisting of repeating Ser-Gly residues and cleavage sites for two restriction enzymes, BamHI and SalI. These sites have been positioned to be able to insert short sequences of interest. In this open flexible linker region we can insert different sets of sequences, e.g., suitable for the analysis of tryptic enzymes, chymotryptic enzymes or elastases, respectively. In this way, we can analyze enzymes with any conceivable primary specificity. We have so far produced more than 370 such substrates to finely map the specificity of the chymase-locus-encoded proteases described here, as well as those of other hematopoietic and non-hematopoietic proteases, including thrombin and the duodenases [11,32,33,34,35,36,37]. A schematic drawing of this type of construct is also shown in an earlier publication [14].

To enhance the understanding of the specificity during cleavage we will give a short description of the nomenclature concerning the region of the cleavage site. When a peptide bond is cleaved (hydrolyzed), the resulting amino acid found C-terminally of the broken bond is denoted P1′, P2′, etc., and the N-terminal amino acid residues are termed P1, P2, etc. (Figure 3A) [38]. The substrate P-sites interact with their counterpart sub-sites in the enzyme, which are denoted S1, S2, S1′, S2′, etc. Cleavage of the peptide bond always occurs between P1 and P1′ in the substrate (Figure 3A).

Proteases can be classified based on their preference for cleaving certain types of amino acids in the P1 position of the substrate. For example, enzymes that cleave after aromatic amino acids (Phe, Tyr and Trp) are generally classified as having chymotrypsin-like activity. These enzymes are among the hematopoietic serine proteases usually named chymases or stated to have ‘chymase activity’. Enzymes cleaving after basic positively charged amino acids such as arginine and lysine are instead classified as having trypsin-like activity and among hematopoietic serine proteases named tryptases (having ‘tryptase activity’). Moreover, enzymes that cleave after small aliphatic amino acids are named elastases (having ‘elastase activity’), enzymes that cleave after leucine are named leu-ases, enzymes that cleave after methionine are named met-ases, and finally enzymes that cleave after negatively charged amino acids such as aspartic acid or glutamic acids are called Asp-ases. However, there are exceptions to this rule, often for historical reasons. For example, cathepsin G is a chymase, whereas mMCP-5 and rMCP-5 are classified as α-chymases based on sequence homology to the human chymase, but they are elastases, due to mutations in the S1 pocket.

Using phage display, we have determined the extended cleavage specificity of a number of the chymase-locus-encoded proteases. The extended specificity is a more detailed description of the specificity of a protease, also involving residues surrounding the cleavage site, often 4–5 amino acids N- and C-terminally of the actual cleavage site. Using phage display, the extended specificity has been determined for the human α-chymase, human cathepsin G, and the α-chymases of macaques and dogs [11,17,32,33,39]. A very similar extended specificity was found for all these proteases (Figure 4). They all show a preference for the aromatic amino acids Phe and Tyr over Trp and Leu in the P1 position. They also show a preference for negatively charged residues in the P2′ position and for aliphatic amino acids in most other positions both N- and C-terminally of the cleavage site (Figure 4). The two most notable differences between these proteases are, firstly, that the dog chymase shows a slightly lower preference for negatively charged residues in the P2′ position compared to the human and macaque chymases and, secondly, that human cathepsin G shows a much lower overall chymase activity compared to the mast cell chymases. Human cathepsin G also shows substantial activity against the basic amino acid Lys, demonstrating that it is a dual enzyme, having both chymase and tryptase activity [17]. Cathepsin G therefore has specificity for both aromatic amino acids (Phe and Tyr) and for one of the two basic amino acids (Lys) (Figure 4). To look closer into the structural requirements for the P2′ selectivity for negatively charged residues, we examined the structure of the human chymase and identified two positively charged amino acids (Arg143 and Lys192), which seemed to be positioned at or close to the position of the negatively charged P2′ residues (in the substrate) during a cleavage reaction. We produced single-amino-acid mutants of the human chymase for each of the two positively charged residues and also a double mutant, changing both of them to non-charged residues [33]. These three mutants were then analyzed by means of phage display and we could see that both of the single mutants reduced the number of negatively charged residues in the phages obtained in the phage display by approximately 50% (Figure 5) [33]. The double mutant showed no selection for negatively charged residues over the background, suggesting that both of these two residues contributed almost equally to this P2′ selectivity for negatively charged residues (Figure 5) [33]. Interestingly, we also found that the activity of several chymase inhibitors was markedly affected by the altered P2′ selectivity of the mutants, indicating that structural information concerning the extended specificity of a protease can be used to construct highly specific protease inhibitors [40].

The specificity of a large number of the rodent chymase-locus-encoded enzymes have also been determined via phage display. The specificity of both α-and β-chymases from rats, mice and golden hamsters has been analyzed in detail [24,41,43,44,45]. It is interesting to note that all of the α-chymases (Cma1), in mice, rats and hamsters have changed their primary specificity from exhibiting chymase activity to now having elastase activity with selectivity for aliphatic amino acids such as Val, Ile and Ala (Figure 6) [24,44,46,47]. These are, for historical reasons, still being named α-chymases. Interestingly, although the α-chymases have lost their chymotrypsin-like activity, some of the β-chymases have apparently taken over the role of the α-chymases (Figure 4 and Figure 6). These β-chymases include the mouse mMCP-4, the rat rMCP-1 and the hamster chymase II, all showing primary and extended cleavage specificity almost identical to the single human mast cell chymase, an α-chymase (Figure 4) [24,41]. Therefore, these β-chymases are considered to be the closest functional homologs to the human chymase and they are also primarily expressed in CTMCs. In addition to these β-chymases, rodents also have additional β-chymases, of which most are expressed by MMCs, and not by CTMCs. Members of this category are the mouse mMCP-1 and the rat rMCP-2, rMCP-3 and rMCP-4 (Figure 6) [36,43,45]. In the rat there is also a β-chymase that is not expressed in hematopoietic cells but in vascular smooth muscle cells, which is therefore named the rat vascular chymase (RVC) [48]. All of these MMC-specific, or blood vessel wall (RVC)-specific β-chymases have chymotrypsin-like activity, showing a preference for aromatic amino acids in the P1 position, and similarly to the primate α-chymases they also show a preference for aliphatic amino acids at several positions, both N- and C-terminally of the cleavage site. However, they lack the preference for negatively charged amino acids in the P2′ position that is characteristic of the primate α-chymases (Figure 6 and Figure 7) [36,43,45,48] and furthermore, they have preferences for Ser or Arg in the P1′ position, just C-terminally of the cleavage site, which is not seen in primate α-chymases (Figure 4, Figure 6 and Figure 7) [36,43,45].

Interestingly when analyzing the β-chymase in rabbits and the α-chymase in guinea pigs (Cma1-like and Cma1), we can see that they have both become strict Leu-ases and are no longer classical chymases (Figure 7) [49,50]. To date, these two enzymes are the only ones of all the studied hematopoietic serine proteases that exhibit specificity for Leu in the P1 position. The biological consequence of this change in specificity is not yet known. However, the fact that a β-chymase in the rabbit, but an α-chymase in the guinea pig, have both become Leu-ases with very similar extended specificities, suggests an independent change of specificity in these two species (Figure 1) [49]. Notably, the α-chymase in the rabbit and the β-chymase in the guinea pig have been inactivated and become pseudogenes.

Sheep have one α-chymase but no β-chymase (Figure 1). Analysis of the extended cleavage specificity via phage display shows a cleavage preference that is similar but not identical to that of the human chymase. The sheep enzyme shows a preference for Phe, Tyr and Trp over Leu in the P1 position and for aliphatic amino acids both N- and C-terminally of the cleavage site [51]. However, in contrast to the human and macaque chymases, mMCP-4, and hamster and opossum chymases, the sheep chymase show no or very low preference for negatively charged residues in the P2′ position [51]. Together with the rat chymase (rMCP1), the sheep chymase (MCP2) thereby forms a small subfamily of mammalian chymases that show fairly unspecific preferences in the P2′ position (Figure 4) [51].

In the opossum there are only two enzymes encoded in the chymase locus, one chymase and one granzyme B homologue. Upon analysis of the opossum chymase via phage display we found that it has a similar specificity as the human α-chymase; however, it has one major difference, seen in the P1 position of the substrate, where the opossum enzyme prefers Trp over Phe and Tyr, whereas the human enzyme prefers Phe and Tyr (Figure 4) [42]. Using a number of recombinant substrates, we found that the opossum granzyme B is a true Asp-ase, as is the case also for the human and mouse counterparts [52].

All three chymase-encoded enzymes in the platypus have been studied by the use of a panel of recombinant substrates. Several attempts with phage displays did not produce sufficient amplification during the panning, for unknown reasons, which is why the recombinant substrates were used as an alternative method to obtain information on their extended specificity. We found that two of the platypus enzymes are chymases, named granzyme B and DDN-1-like, and exhibit very similar specificity to that of the human chymase. They both show preferences for Phe and Tyr over Trp and Leu in the P1 position and also show preferences for aliphatic amino acids both N- and C-terminally of the cleavage site [53]. The third platypus enzyme, granzyme BGH-like, is located on another contig (Figure 1). This enzyme was found to be a classical Asp-ase with a preference for negatively charged residues in the P1 position, similarly to human, rat and opossum granzyme B [52,53,54].

We do not yet have any information on the specificity of the reptile, bird and amphibian enzymes of the chymase locus even though we have produced three of them as recombinant enzymes: Xenopus CTSG and GzmH, and Chinese alligator MCP1A-like enzymes. Several attempts, using either phage displays or chromogenic and recombinant substrates, have not resulted in any information of value, which is why more detailed information concerning the extended specificity of these enzymes has only been obtained for their mammalian counterparts.

The P1 specificity of the mammalian enzymes can be predicted based on the fact that enzyme specificity amongst the members of the large family of chymotrypsin-related serine proteases has been found to strongly correlate with the substrate binding pocket (the S1 pocket), where residues 189, 216 and 226, according to chymotrypsinogen numbering, play a key role. These residues are highly complementary to the substrates, determining the primary specificity (see Figure 3B). For example, the negative Asp189 in the S1 pocket accommodates positively charged Arg or Lys in the substrate, in the case of trypsin. The steric properties of certain amino acids are also important, for example, in elastases, slightly larger, usually non-polar amino acids create a smaller S1 pocket [55]. Hence, only small amino acids can be accommodated in the elastase S1 pocket (Figure 3B). It should be noted that these three residues (189, 216 and 226) are not the only determinants of the enzymes’ primary cleavage specificity, but they can often provide a good prediction of the primary specificity of an enzyme. By looking at these three residues, we found that Xenopus CTSG has the same amino acid combination as the granzyme B proteases of placental mammals, indicating that it is a granzyme B-like enzyme. However, we have not detected any cleavage of a panel of 2xTrx Asp-ase substrates by this enzyme, indicating that it either has a markedly different extended specificity, or that it is much more specific, compared to mouse, human and opossum granzyme B. With more specificity, we refer to that that the enzyme is sensitive to the type of amino acid in several positions surrounding the cleavage site, and not only to the P1 position. When looking at related enzymes in fish in other chromosomal loci, these three residues have in our hands not provided any reliable information concerning this expected primary specificity. The sequence divergence is too large to specify the location of a particular amino acid forming the S1 pocket based only on sequence alignment with the mammalian enzymes [19].

In a recent study we have looked into the new subfamily of chymase-locus-encoded proteases in ruminants and pigs, the duodenases [31]. These proteases are most likely a result of a set of gene duplications originating from a cathepsin G or a granzyme gene (Figure 1). Six of the duodenases were analyzed through either phage displays or recombinant substrates and we found that they showed a surprisingly broad panel of different specificities (Figure 8). One of the analyzed duodenases was a highly specific asp-ase (bovine MCP1A); two were highly specific tryptases (sheep MCP3 and MCP3-like); two enzymes exhibited dual specificity, being both tryptases and chymases (bovine DDN1 and DDN1-like); and one enzyme was a pure chymase (pig MCP3-like) with a relatively broad extended specificity (Figure 8) [31]. Notably, these six duodenases are all highly homologous in their primary sequence despite their large differences in primary and extended specificity, illustrating that the enzyme specificity can be changed quite dramatically by only minor changes in the primary sequence.

## 4. Phylogenetic Analysis

To study the relatedness in sequence between the different enzymes encoded from the *chymase locus*, we have performed a larger phylogenic analysis of the majority of the *chymase locus* genes included in Figure 1 and Figure 2 and presented the result in the form of a phylogenetic tree (Figure 9). For all proteases, the entire sequence of the active form, not including the signal sequence and activation peptide, were used in the multiple alignments. The phylogenetic analyses were performed using a Bayesian approach as implemented in MrBayes version 3.2.7a, and the phylogenetic tree was drawn in FigTree 1.4.2.

All the chymase locus genes form a larger branch, separated from enzymes of the other hematopoietic serine protease loci, including the genes within the met-ase, the T cell tryptase and the mast cell tryptase loci (Figure 9A). The separate bird and reptile locus is also found as a separate subbranch (Figure 2 and Figure 9) [19]. The genes within the mast cell tryptase locus are the most distantly related members of the different hematopoietic serine proteases, indicating that they may not have been part of the same tetraploidization process that may have formed the other loci-encoding hematopoietic serine proteases [19,56]. In addition, all the α-chymases, the β-chymases, the cathepsin G genes, the granzymes, the mMCP-8 members and the duodenases form separate subbranches (Figure 9B). We can also see that both the mMCP-8-like and the duodenase subbranches are positioned closer to the granzymes and cathepsin G than to the chymases indicating that both the mMCP-8 related enzymes and the duodenases originate from duplications of a granzyme or cathepsin G gene (Figure 1 and Figure 9).

## 5. Sequence Alignment of Reptile and Amphibian *Chymase-Locus*-Encoded Proteases

To obtain clues as to the primary specificity of the still uncharacterized reptile and amphibian *chymase-locus*-encoded enzymes, we have aligned their amino acid sequences and looked at the three residues (in positions 189, 216 and 226) forming the S1 pocket (Figure 10). As can be seen in the figure, human and mouse granzyme B, which are used as reference proteases with asp-ase activity, have the triplets TGR and AGR, respectively, whereas human Cma1 and human granzyme H, which both are chymases, have the triplets SGA and TGG, respectively (Figure 10). Due to the presence of a positively charged arginine residue in position 226 (i.e., the R in TGR and AGR), human and mouse granzyme B do prefer negatively charged amino acids in the P1 position of the substrate and thereby they are asp-ases. Using the same criteria, we can see that the Xenopus enzymes (the two AC and one WC frog sequences), which all have SGR triplets, are most likely asp-ases and thereby granzyme B homologs (Figure 10). Similarly, the Chinese alligator MCP1A-2, with an SGK triplet, and Gopherus turtle GzmBL, and MCP1AL-1 and 2, with AGR triplets, also seem to be granzyme B homologs. These findings indicate that granzyme B already appeared in amphibians and is represented in both reptiles and mammals, but seems to have been lost in birds. In the turtle, we can also find several proteases with SGS or AGS triplets, indicating that they are classical chymases due to the S in position 226 (Figure 10). However, we also find one enzyme with an LVS triplet, the G-Turtle CtsGL, indicating that it may be an elastase, as the presence of both a Leu and a Val residue may limit the space in the pocket so that large aromatic amino acids may not be able to enter (Figure 10). Then we have two additional Chinese alligator proteases, MCP1A-1 and MCP1A-3, with SGD and SGE triplets and one American alligator sequence with SGE (Figure 10). The presence of a negatively charged residue in position 226 may indicate that these proteases prefer positively charged amino acids in the P1 position of substrates. However, a similar situation is seen for primate cathepsin G, which has dual specificity with both chymase and tryptase activity. For example, human cathepsin G (with triplet AGE) is both a classical chymase with a preference for large aromatic amino acids such as Phe and Tyr, but it also accepts Lys in the P1 position [17]. Based on the similarity in the triplets of the S1 pocket, we may speculate that the three alligator proteases could also display such dual activity.

## 6. Discussion

The chymase locus has experienced quite dramatic changes during tetrapod evolution, and an interesting question is how these changes reflect the need for new functions related to the life conditions of the separate tetrapod lineages. The massive increase in the number of genes within this locus in several rodent species indicates a need for additional mast cell proteases and also for additional granzymes in rodents. However, the functional significance of these massive increases has not been easy to identify. Why would rats and mice need 18 and 15 functional genes, respectively, when humans and other primates apparently only need four such genes? A feasible explanation could be that these additional proteases in rodents have provided an advantage for survival in the more pathogen-rich rodent habitat. Although there is still no direct proof for such a beneficial role for many of these additional proteases, this notion is well in line with the recently reported functions attributed to some of these proteases (described in more detail below). Moreover, the importance of these additional proteases is strongly indicated by the seemingly independent amplifications of these genes in several rodents, including mice, rats, hamsters and guinea pigs. The large difference in the numbers of genes and positions of genes within the chymase, mMCP-8-like, cathepsin G and granzyme subfamilies strongly indicate independent duplication processes. On the other hand, the opossum, a marsupial with a very similar lifestyle to that of rodents, have only two chymase-locus-encoded genes, the mast cell chymase and granzyme B. The apparent question is then why the opossum, with a similar lifestyle to that of rodents, can manage with only two chymase-locus-encoded serine protease genes. We have some indications by PCR analysis of skin and intestinal tissue samples that gene amplifications may have occurred in other serine protease loci to compensate for the low number of proteases in the chymase locus of the opossum. However, this indication needs to be further analyzed before we can make any more definitive conclusions. In the opossum skin samples, we have found PCR fragments for proteases related to pancreatic digestive enzymes. Interestingly, an apparent expression of pancreatic elastase has been demonstrated in human skin tissue, indicating a similar process in primates, which also have fewer chymase locus genes [57]. The horse is also an example of a species in which a massive expansion has occurred in one part of the locus, the granzymes. Although ruminants share a similar lifestyle with that of horses, they do not exhibit such a massive increase in the number of granzymes genes (Figure 1). Altogether these findings show that there are major differences within different parts of the mammalian evolutionary tree in the extent of gene amplifications we see in the chymase locus.

A crucial step in uncovering the beneficial role of these evolutionary “new” genes of the chymase locus will be to identify their true biological targets. For example, we found that the new mucosal mast cell specific β-chymases, mMCP-1 in mice and rMCP-2 in rats, can cleave cell adhesion proteins, and so they are most likely involved in the opening of the intestinal mucosa as part of the defense against infections [36]. The release of these proteases will most likely increase the outflux of protective immune proteins and cells, hence reducing the impact of the intestinal parasite infection [36]. Moreover, the basophil-specific protease mMCP-8 may be involved in the defense against ticks and other skin parasites. This was supported by a study demonstrating that mMCP-8, when injected into the skin of mice, could elicit a local inflammatory response, including tissue swelling with increased microvascular permeability and infiltration of leukocytes, primarily neutrophils [58]. Accordingly, the local release of mMCP-8 in response to skin parasites may have beneficial effects by enhancing the potential of sensing the presence of these parasites and by inducing a potent inflammatory response [58]. It is known that rodents are affected by large numbers of both intestinal helminth infections and also skin parasites, including ticks, lice and flees, and that basophils accumulate at the site of tick feeding in the skin of mice [59]. Together these findings indicate a beneficial role of some of these additional proteases in the defense against both intestinal worm parasites and against ectoparasites of the skin. However, the functional consequences of all the additional granzymes found in rats, mice, hamsters, guinea pigs and horses are almost totally unexplored (Figure 1) [28].

The beneficial effects of the increase in the number of protease genes in the chymase locus of ruminants may be easier to understand. Cattle and sheep have three and five additional functional duodenase genes, respectively. These duodenases are all located within the chymase locus and they all belong to a completely new subfamily of chymase-locus-encoded protease genes [19,60]. These genes most likely originated from duplications of a cathepsin G gene or a granzyme gene. These duodenases are no longer expressed by hematopoietic cells, like almost all other proteases encoded from this locus, but are instead expressed by the Brunner´s glands of the duodenum [23]. This expression pattern has so far only been shown for the bovine duodenases, but is probably also valid for sheep and pigs. Ruminants have a very complex digestive system, which may have become a strong selective force in gene evolution. A very interesting example of this force is seen in another locus related to immunity, namely, the classical lysozyme (lysozyme C) genes. Lysozyme C is involved in cleavage of the peptidoglycan of the bacterial cell wall. In the human genome there is only one classical lysozyme gene. In sharp contrast, cattle have ten such genes and at least four of them are expressed by the abomasum, the real stomach [61,62]. The lysozyme in the abomasum participates in food digestion by breaking down the cell wall of bacteria entering the stomach from the other parts of the digestive system. The bacteria can thereby more efficiently be used as a food source. This process is particularly important considering that ruminants rely on bacteria and other microbes to transform the ingested cellulose from plants into digestible macromolecules. The lysozyme genes in cattle represent one example of how gene amplification has facilitated the adaptation to a new way of life and in this case a new food source. The new genes in the chymase locus coding for duodenases may also have emerged as a consequence of the unique digestive system in ruminants. However, the role that these duodenases play in the digestion of food for ruminants is largely unexplored. We have recently shown that these duodenases, which are highly homologous in sequence, have become enzymes with very diverse both primary and extended cleavage specificities, indicating several different biological targets [31]. Based on a screening of the total proteome of cattle or sheep using the consensus sequences of the extended cleavage specificity of two of these enzymes, we found that mucin-5B, the major salivary mucin, is one candidate substrate for the bovine enzyme, and several chemokine receptors are candidate substrates for the sheep enzyme. These findings suggest that these two enzymes can be involved in the digestion of salivary mucins and in the dampening of inflammatory processes in the duodenum induced by massive bacterial products entering from the abomasum.

This and earlier studies of the chymase locus have shown that a large number of gene amplifications and subsequent diversification in cleavage specificity have occurred among these enzymes during tetrapod evolution, showing that this locus is actively responding to changes in lifestyle and environment. Although large differences between species are seen in the overall pattern of this locus, the mast cell chymase and granzyme B are two genes that seem to be highly conserved, especially during mammalian evolution. These enzymes exhibit similar specificities in all studied mammalian species from platypus to humans, with two exceptions being rabbits and guinea pigs (Figure 7) [49]. In these two latter species, the chymases have become highly specific Leu-ases and have lost their chymotrypsin-like activity. The question is then if rabbits and guinea pigs have enzymes from other loci that exhibit chymase activity and are expressed by mast cells, or if they have developed other strategies to compensate for the lack of the critical functions exerted by the highly conserved chymases? No detailed analysis of mast cells from these two species has yet been presented that could give clues as to this intriguing question.

Another topic of interest is the presence of two mMCP-8 like genes in the horse that cluster with the rodent mMCP-8-like genes in the phylogenetic tree (Figure 9). This is the first example of mMCP-8-like genes outside of the rodent lineage and it highlights the question of the tissue specificity of these two new genes. Are they basophil specific like the mouse mMCP-8, or are they also expressed by mucosal mast cells as is observed in the rat [26]? The identification of these mMCP-8-like genes in the horse and the β-chymase in cats also show that both β-chymases and the mMCP-8 subfamilies are not purely rodent-specific, as previously assumed, but they seem to have appeared earlier during mammalian evolution and may have been lost in several other mammalian lineages. A broader analysis of additional mammalian lineages may give a more detailed picture of their origin, as well as the timing of their appearance.

Detailed analyses of the extended cleavage specificities of a large number of the chymase-locus-encoded proteases from different species have resulted in initial tools to predict the in vivo substrates of these proteases. However, sensitivity to folding and thereby accessibility to potential cleavage sites also play a major role in determining which targets will be cleaved or not. A more detailed analysis of the cleavage of potential targets is therefore needed before we can say with certainty that they are likely targets. This was particularly evident during the analysis of the cleavage by the human mast cell chymase and human cathepsin G of a large panel of cytokines and chemokines [15]. Only three cytokines and two chemokines were cleaved by the human chymase out of a panel of 55 cytokines and chemokines analyzed [15]. Consensus sites that were hidden or partly hidden in the structure were not cleaved, whereas suboptimal sites that were highly exposed could serve as relatively good targets for the enzyme, which shows the high complexity in this type of studies. Phage display is very efficient in identifying the most optimal cleavage site in a linear sequence. However, sites that are only 2–4 times less favored by the enzyme can then often be overlooked and thus not identified by the phage display, but these sites can still be biologically relevant. One such example is human cathepsin G, for which the cleavage of Lys-containing substrates was not detected through the phage display [17]. However, this cleavage was found when using recombinant Lys-containing substrates and the rate of cleavage was only 2–3 times lower compared to the cleavage of substrates containing the most preferred residues, Phe and Tyr [17]. Further analysis of the most important targets for all of the chymase-locus-encoded proteases are therefore needed before we can fully understand the role of these proteases in tetrapod immunity, in addition to understanding what forces have shaped the locus during more than 400 million years of tetrapod evolution. The question of the origin of this locus is also highly interesting, especially since no clear identifiable ancestral locus can be found in any of the fish species studied. How have fishes and birds managed to perform the tasks carried out by the chymase locus proteases? It is possible that a convergent evolution of proteases from other loci provide these functions in fishes and birds. An indication of such a process was found when analyzing a protease, catfish I, expressed by catfish NK-like cells [37]. This fish protease was found to be highly specific and caspase 6 was identified as one likely target indicating that this protease, encoded from another locus, may be involved in inducing apoptosis in virus-infected cells, similarly to granzyme B in mammals [37].

By analyzing the three amino acids forming the S1 pocket of the serine proteases encoded from the chymase locus of amphibians and reptiles, we can see that a number of these enzymes have triplets, indicating that they are true granzyme B homologs. Such potential homologs were found in four out of the five species studied, the two frog species, the Chinese alligator and the G-Turtle (Figure 10). This strongly indicates that granzyme B may have been the first enzyme to appear in this locus, possibly during early tetrapod evolution, and that the true chymases and cathepsin G homologs did appear slightly later during the early reptile evolution. These assumptions are merely based on the S1-pocket-forming triplets, which have relatively good predictive value, and on the fact that no chymase or cathepsin G homolog is to be found in the two frog species. However, for a more detailed view of the specificity of these enzymes we need to determine their extended specificity, and this project is ongoing, but has been found to be relatively challenging. We have made several attempts but have not succeeded with either phage display or recombinant substrates, indicating that these amphibian and reptile enzymes may have higher selectivity than the mammalian enzymes and may therefore be more difficult to study using the abovementioned methods. Such indications have also come from studies of related enzymes from other loci in fish. The catfish I enzyme is presently the most specific hematopoietic serine protease that we have studied [37]. It was found to be a highly specific met-ase with selectivity for the majority of amino acid positions at and surrounding the cleavage site over a region of 8–9 amino acids. There is a possibility that these amphibian and reptile enzymes are particularly difficult to study, due to their high specificity. However, the success with catfish I shows that this is not impossible, and there is a good chance that in the near future we will also obtain confirming data on the extended specificity of these amphibian and reptile enzymes.

## Figures and Tables

**Figure 1 ijms-22-10975-f001:**
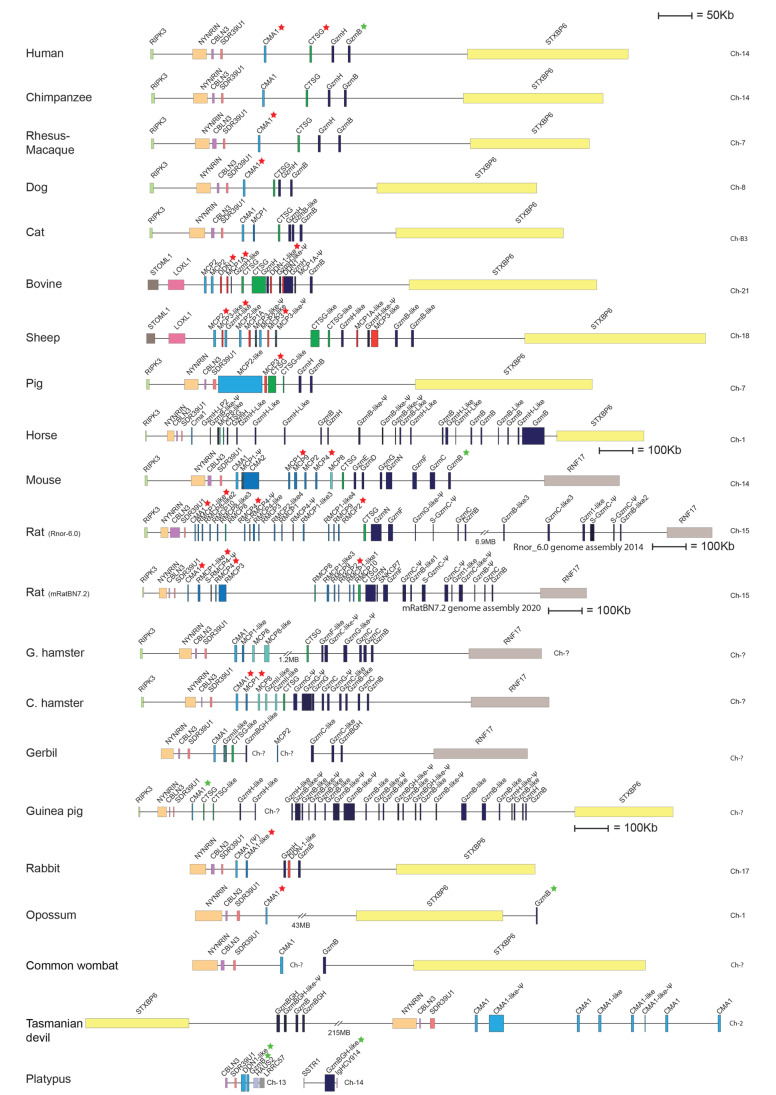
The *chymase locus* of a panel of mammalian species. An in scale figure of a panel of mammalian species ranging from humans to marsupials and monotremes. The genes are color coded. Granzymes are shown in dark blue, α-chymases in light blue and β-chymases as blue with a darker tint. mMCP-8-related genes are in cyan, duodenases in red and cathepsin G in green. The general scale bar of 50 kB is valid for most of the genes. However, three loci are larger in size why we have reduced the size to half. For these three species, horse, rat and guinea pig, we have introduced a 100 kB size marker. All the genes for the proteases analyzed by phage display are marked with a red star and the ones analyzed by a panel of recombinant substrates by green stars. The figures were made in Adobe Illustrator and is a heavily updated version of figure 11 in Thorpe et al. 2018 [24].

**Figure 2 ijms-22-10975-f002:**
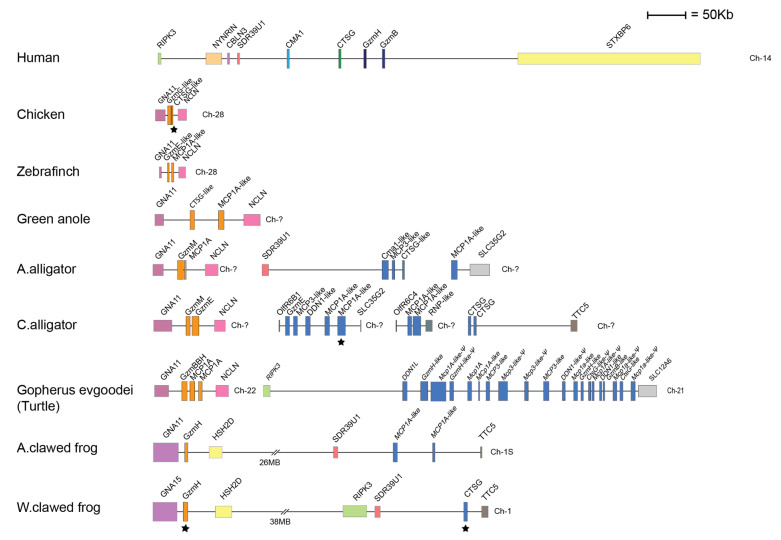
The *chymase locus* and a related locus of a panel of non-mammalian tetrapod species. An in scale figure of a panel of non-mammalian tetrapods ranging from amphibians to birds. The genes are color coded. In the human locus the granzymes are shown in dark blue, α-chymases in light blue, β-chymases as blue with a darker tint and cathepsin G in green. In the amphibians and reptiles the genes cannot be assigned to any of the major groups of mammalian enzymes and are therefore shown in a light-dark blue color. The genes for which we have produced recombinant proteins are marked by black stars. The protease genes within the additional locus present in birds, reptiles and amphibians but has been lost in mammals are shown in orange.

**Figure 3 ijms-22-10975-f003:**
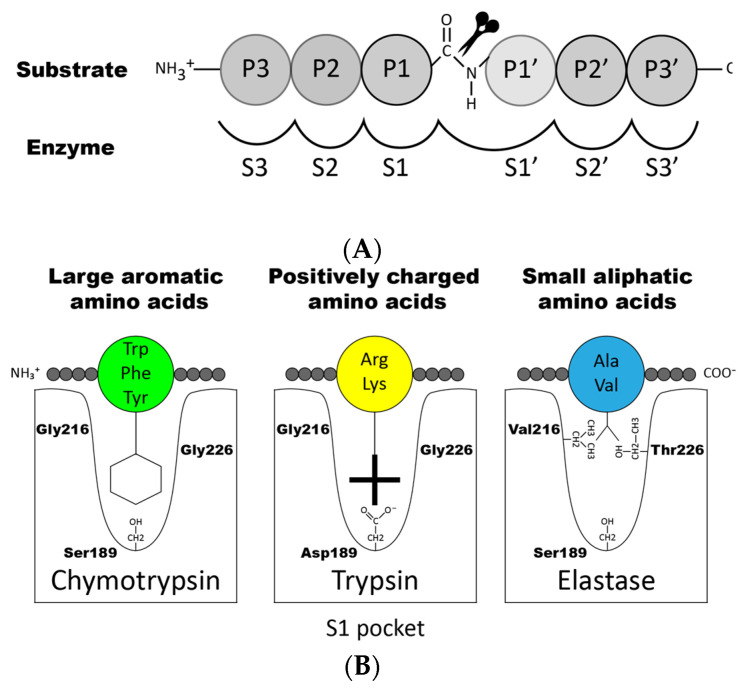
Nomenclature surrounding the cleaved peptide bond. (**A**) shows the amino acids N-terminal from the cleaved bond that are termed P1 (where cleavage occurs, depicted by scissors), P2, P3 etc. Amino acids C-terminal of the cleaved bond are termed P1′ (adjacent to P1), P2′, P3′ etc. The corresponding interacting sub-sites in the enzyme are denoted with S. (**B**) shows the S1 pocket, which is important in determining the primary specificity of the chymotrypsin family. The important residues are shown, which determine chymotrypsin-, trypsin- or elastase-like specificity. The figure is a modified and combined version of figures 6 and 7B from Hellman and Thorpe 2014 [14].

**Figure 4 ijms-22-10975-f004:**
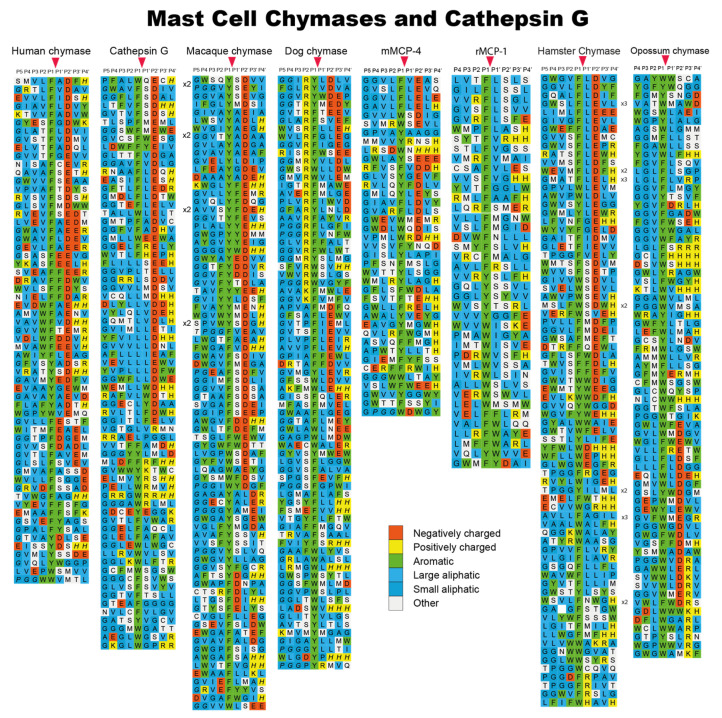
Phage display results for a panel of mammalian connective tissue mast cell chymases and of human cathepsin G. Phage display results for an analysis of the human mast cell chymase, the human cathepsin G, the mast cell chymase of the macaque, the dog chymase, mMCP-4, rMCP-1, hamster mast cell chymase and the opossum mast cell chymase. The amino acids are color coded as shown in the bottom of the figure. The cleavage by the enzyme occur after the P1 residue as marked by red arrow heads. The figure contains data from six earlier publications [11,24,32,39,41,42].

**Figure 5 ijms-22-10975-f005:**
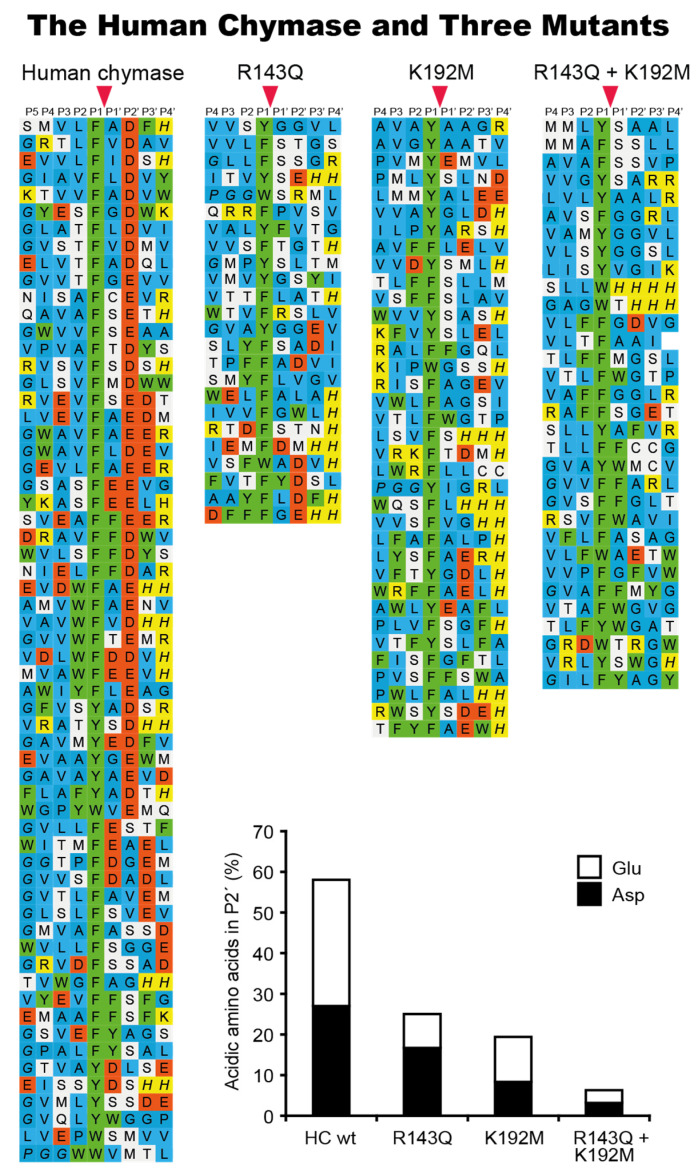
Phage display results for the analysis of the importance of Arg143 and Lys192 in the human chymase for the selection of negatively charged amino acids in the P2′ position of substrates. In the bottom of the figure a diagram of the percentage of negatively charged amino acids found in the P2′ position of different phages isolated from the phage display with the wild type enzyme, the two single mutants and the double mutant. The relative distribution between the two negatively charged amino acids, Asp and Glu, is also shown by black or white fields. The amino acids in the phage display results are color coded as shown in Figure 4. The cleavage by the enzyme occur after the P1 residue as marked by red arrow heads. The figure contains data from two earlier publications [33,39].

**Figure 6 ijms-22-10975-f006:**
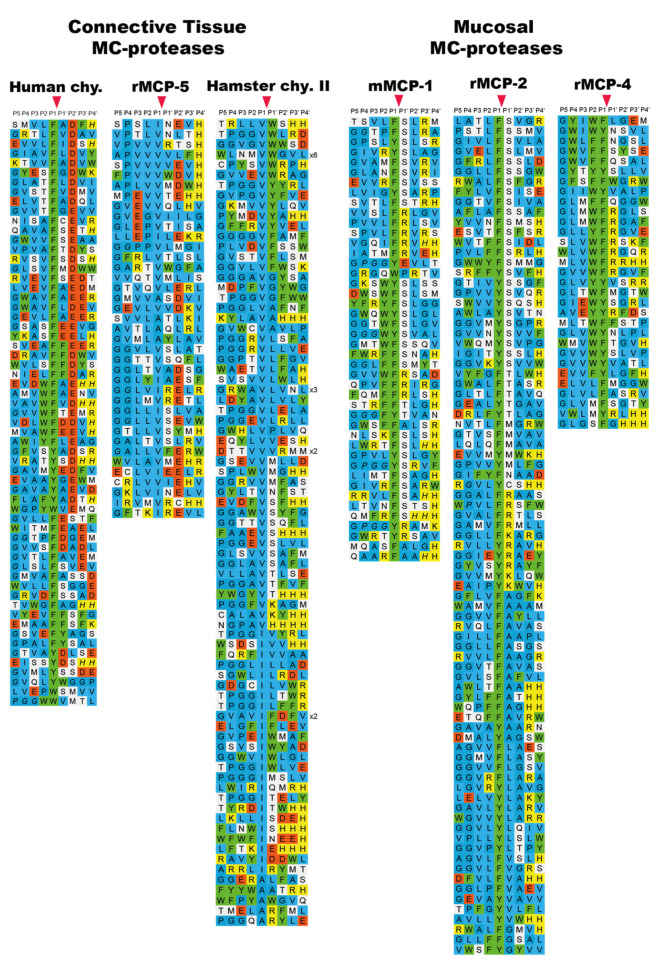
Phage display results for a panel of rodent α-chymases that has become elastases and of a panel of mucosal mast cell specific β-chymases. The amino acids in the phage display results are color coded as shown in Figure 4. The cleavage by the enzyme occur after the P1 residue as marked by red arrow heads. The figure contains data from six earlier publications [35,38,40,41,42,44].

**Figure 7 ijms-22-10975-f007:**
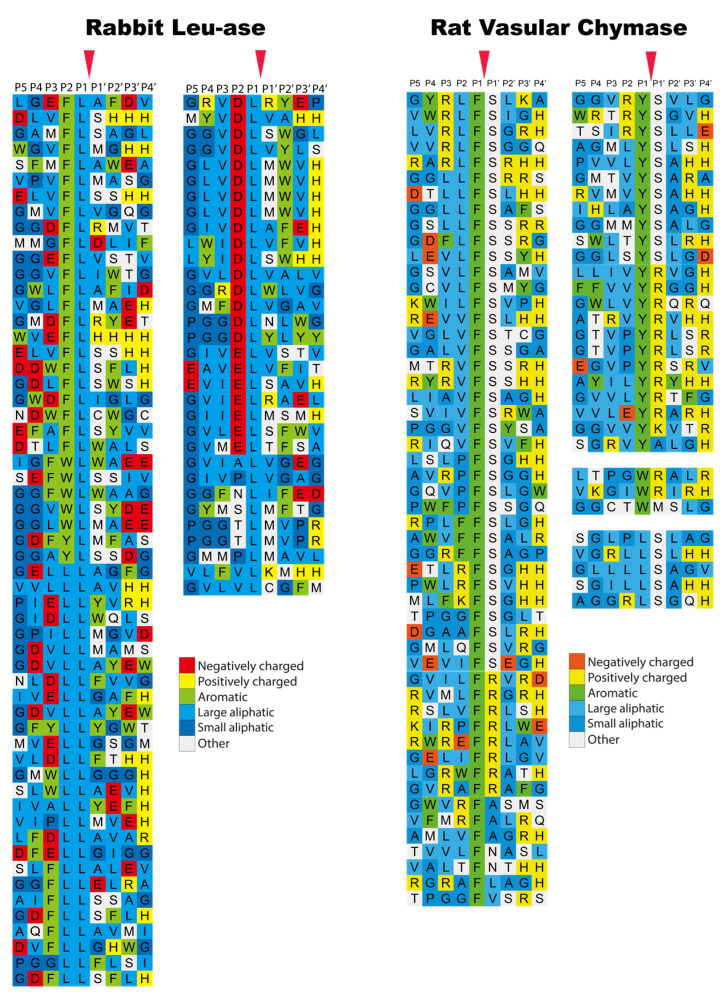
Phage display results for the rabbit β-chymase, a Leu-ase, and the rat vascular chymase (RVC). The amino acids in the phage display results are color coded as shown in the bottom of the figure. The cleavage by the enzyme occur after the P1 residue as marked by red arrow heads. The figure contains data from two earlier publications [48,49].

**Figure 8 ijms-22-10975-f008:**
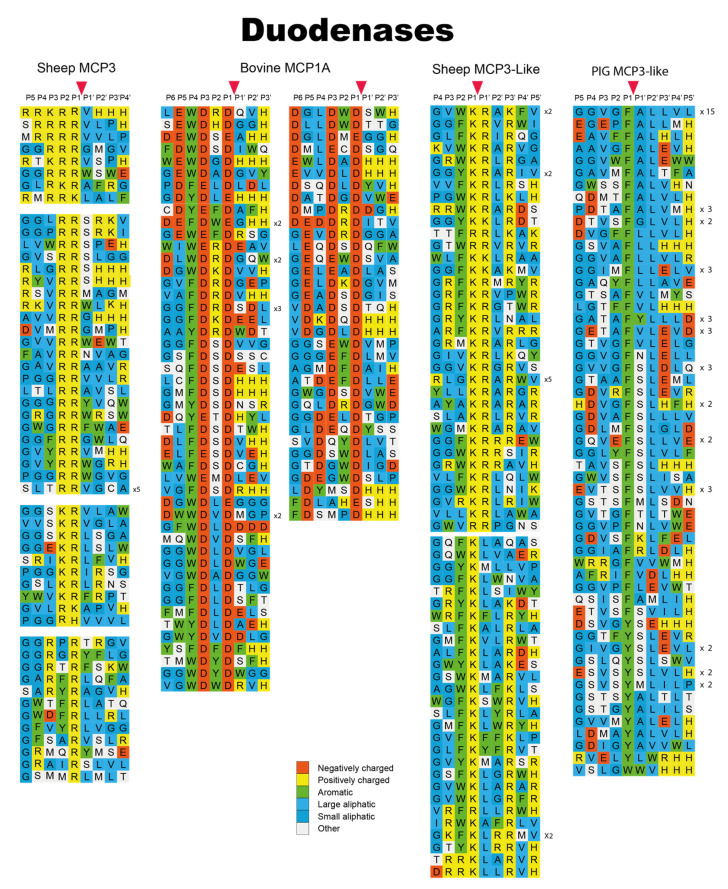
Phage display results for a panel of duodenases from cattle, sheep and pig. The amino acids in the phage display results are color coded as shown in the bottom of the figure. The cleavage by the enzyme occur after the P1 residue. The figure contains data from one earlier publication [31].

**Figure 9 ijms-22-10975-f009:**
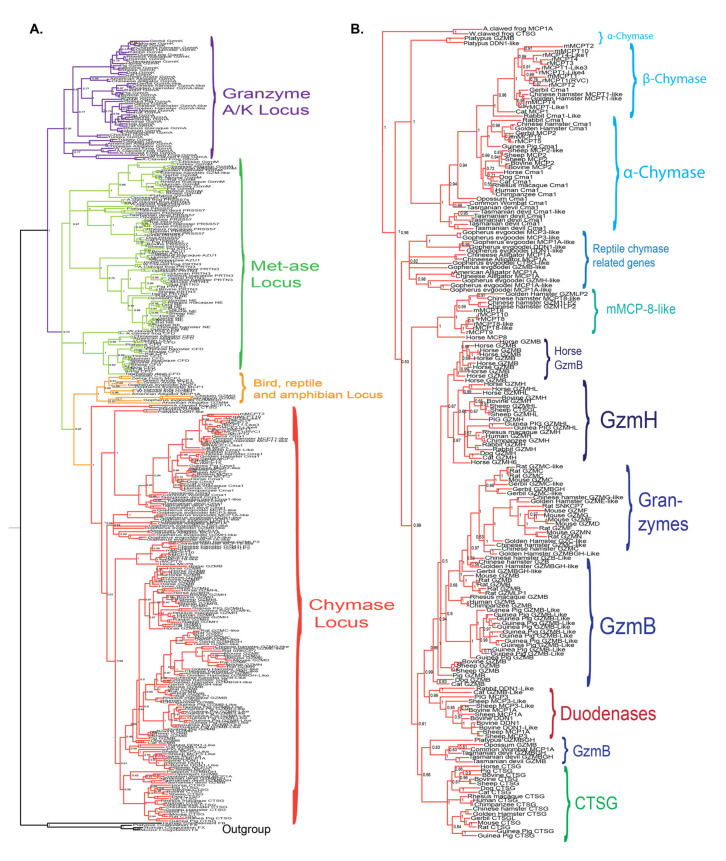
Phylogenetic analysis of a panel of hematopoietic serine proteases with the focus on *chymase-locus*-encoded genes. The amino acid sequences of the active protease, lacking a signal sequence and activation peptide, was analyzed using the MrBayes computer program to generate the phylogenetic tree. The resulting values were then transferred to the FigTree program to generating a tree suitable for figure generation. The results were finally transferred to Adobe illustrator for refining of the figure and the addition of explanatory text and decorations. The different branches of the tree have been color-coded for a clearer visualization of individual branches. A few non-hematopoietic serine proteases, pancreatic trypsin and blood coagulation enzymes, were included as outgroups to help form a more stable tree. (**A**) shows a tree including proteases from four different loci to put the *chymase locus* in the context of the other loci encoding hematopoietic serine proteases. (**B**) shows an enlarged picture of the *chymase locus* genes of panel A to be able to see the names of the individual genes.

**Figure 10 ijms-22-10975-f010:**
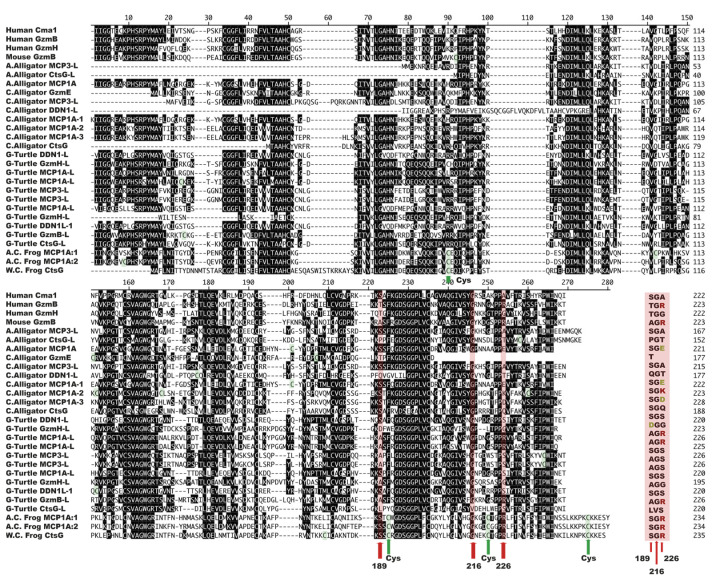
Alignment of the chymase-locus-encoded genes of Figure 2 from reptiles and amphibians. The amino acid sequences of the active protease, lacking a signal sequence and activation peptide, were aligned using the DNASTAR program to visualize the residues forming the S1 pocket of the enzyme. The three amino acids forming the S1 pocket, which can give important information concerning the primary specificity of the enzyme, are marked by red arrows and summarized as the resulting triplet at the end of the figure, marked in pale red. The three residues forming the S1 pocket are residue 189, 216 and 226, following the position numbering of bovine chymotrypsin [38].

## Data Availability

Not applicable.

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
