# Peer review of "The Evolutionary History of the Chymase Locus -a Locus Encoding Several of the Major Hematopoietic Serine Proteases"

_ijms, 2021, doi:10.3390/ijms222010975_

Round 1

Reviewer 1 Report

Dear Authors,

Proteases of the hematopoietic cell lineages are involved in many protective and detrimental effects, and therefore are promising targets for therapy. Your article “The evolutionary history of the chymase locus -a locus encoding several of the major hematopoietic serine proteases” is interesting and may have impact on this area of research.

  1. The main limitation of this work is the presence of a large number of text borrowings from other works of the authors. For example, the first paragraph of the Introduction section contains many borrowings of whole sentences.

Below are examples of borrowing:

In mast cells, these proteases can account for up to 35% of the total cellular protein, and the absolute majority of these belong to the chymotrypsin-related serine protease family (6).

In mammals, serine proteases belonging to this family have been identified in mast cells, basophils, neutrophils, cytotoxic T cells and natural killer (NK) cells but not in B cells or dendritic cells and only at relatively low levels in eosinophils. A number of very diverse functions have been identified for these proteases, including apoptosis induction, blood pressure regulation, inactivation of insect and snake toxins, killing of bacteria and fungi, mobilization or degradation of cytokines and the degradation of connective tissue components (10-16).

A very broad spectrum of primary cleavage specificities (i.e. chymase, tryptase, asp-ase, elastase and met-ase specificities) has also been observed, which highlights the large flexibility in the active site of these proteases (14). Mast cells (MC) primarily express chymases and tryptases, having chymotryptic and tryptic primary cleavage specificities, respectively. Neutrophils express several enzymes with chymase, elastase and tryptase specificities (14, 17, 18).

T cells and NK cells express between 5 and 17 different granzymes, having tryptase, aspase, chymase and met-ase specificities (14, 19).

The genes encoding these proteases are organized in four different loci: the MC chymase locus, the MC tryptase locus, the metase locus and the T cell tryptase locus, also called the granzyme A/K locus. In mammals, these loci are often located on four different chromosomes, indicating that some of them may originate from whole genome duplications, so called tetraploidizations. Two such tertraploidizations have most likely occurred during early vertebrate evolution, which indicate that the original locus was present before these evolutionary important events (20).”

The excerpts provided were copied by the authors from the following article: Hellman, Lars, and Michael Thorpe. "Granule proteases of hematopoietic cells, a family of versatile inflammatory mediators–an update on their cleavage specificity, in vivo substrates, and evolution." Biological chemistry 395.1 (2014): 15-49.

Another example is interesting. In an article by Hellman, L., & Thorpe, M. Biol Chem (2014), the authors state: “There are 11 different, apparently functional, granzyme genes in this region of the rat genome…”. But in the presented manuscript, the authors indicate: “There are 7 different, apparently functional granzyme genes in this region of the rat genome”. So how many granzyme genes are there?

 IJMS Instructions for Authors states: “Your manuscript should not contain any information that has already been published”. Therefore, this is a violation of the rules of the journal.

  1. Another limitation is that some of the figures (or parts of them) have already been published. But in the manuscript in the captions to the figures, the authors do not indicate from which works the figures were taken.

For example, part of Figure 1 was taken from Figure 11 from Thorpe, M., Fu, Z., Albat, E., Akula, S., de Garavilla, L., Kervinen, J., & Hellman, L. (2018). Extended cleavage specificities of mast cell proteases 1 and 2 from golden hamster: Classical chymase and an elastolytic protease comparable to rat and mouse MCP-5. Plos one, 13(12), e0207826.

Another example, Figure 1 of the manuscript is a combination of Figure 6 and Figure 7B from Hellman, L., & Thorpe, M. Biol Chem (2014). Besides Figure 4, Figure 5, Figure 6 of the manuscript contain parts of figures from 1) Thorpe, Michael, et al. "Extended cleavage specificities of mast cell proteases 1 and 2 from golden hamster: Classical chymase and an elastolytic protease comparable to rat and mouse MCP-5." Plos one 13.12 (2018): e0207826; 2) Thorpe, Michael, et al. "Extended cleavage specificities of mast cell proteases 1 and 2 from golden hamster: Classical chymase and an elastolytic protease comparable to rat and mouse MCP-5." Plos one 13.12 (2018): e0207826.; 3) Akula, Srinivas, et al. "Granule associated serine proteases of hematopoietic cells–an analysis of their appearance and diversification during vertebrate evolution." PLoS One 10.11 (2015): e0143091.

Despite the fact that the listed articles were written by some of the co-authors of the manuscript, please add  references to these articles in the figure captions.

 A significant revision of the article is required.

Best regards

Author Response

Reviewer 1.

  1. The text has now been modified in the introduction so that the information is the same but the wording is different to comply with rules concerning self-plagiarism in IJMS. All the text that have been modified has been marked in red. The language may have deteriorated in quality by these changes but we have now strictly followed the rules.

Concerning the different numbers of functional and non-functional genes primarily in rat is depending on the recent genome update so the new figures confer to the latest update, which we also clearly describe in the text. The highly repetitive regions have made the genome difficult to sequence and annotate in this region why we have been reporting different numbers of active and pseudogenes during the past 10 years as the quality of the rat genome have improved, with every update. We expect the rat genome now to be of reasonable quality so that we can rely on the present figures. By producing recombinant proteins for many of them we have during the past 20 years been able to verify many of them as functional active genes. However, there are a number of genes that we not yet have had time and resources to analyze.  We may manage to do a few more during coming years.

Concerning the statement on published material. I think this is valid for original articles and not for Reviews. Reviews does in general only contain published material but in a new context, like hear, summarizing the work of more than 20 years and more than 20 articles to give an overview of the progress in the field and to be forward looking into future directions.

  1. We have now added references to all the manuscripts where the original data have been obtained from and where we have used and modified whole or parts of previous figures. All figures are modified. Non of them are identical to the original version and the figures are there for the reader to him or herself to analyze the original data in a new context where not only a single protease specificity is presented but a whole family of proteases from up to six different earlier publications in one single figure.

Reviewer 2 Report

In 2006, Prof. Lars Hellman published a review entitled “Regulation of IgE homeostasis, and the identification of potential targets for therapeutic intervention“ (https://pubmed.ncbi.nlm.nih.gov/17145160/). To say that it was excellent is to say nothing. It became a bible to me and many other researchers studying IgE and related things.

Even if the current review turned out te be only half as stimullating as the previous one, it would be a real scientific hit already.

To me personally, this article is already excellent. Full, comprehensive. I have no major reservations.

Minor/technical points:

1. Names of the genes should be always written in italics.

2. Please, check the text carefully, for example the last paragraph of page 2. There are some gaps? Greek letters did not want to convert well?

3. Figures with even higher resolution would be welcome.

Author Response

Reviewer 2.

First I want to give great Thanks for the kind words. It rarely occurs in our world why it is so much more appreciated. Thanks a Lot!!

  1. We have now changed all gene names to italics.
  2. All the Greek letters in symbol have been changed to Times new roman.
  3. I think the quality of the figures become better when not directly transformed from PNG to pdf in the final article as we usually use PNGs as the files are smaller and keeps the color better than TIFF figures, which also are very large and difficult to handle.

Reviewer 3 Report

Srinivas Akula et al. reviewed key concepts collected data concerning the evolution of this rapidly evolving locus, and how these changes in gene numbers and specificities may have affected the immune functions in the various tetrapod species focusing on serine protease.

Points to be considered:

1) The rationale of why the authors came up with this review.

2) What is the information that is not exactly available that motivated the authors to come up with this information. What are the current caveats and how do the authors highlight the current research in answering them? If not, they need to address it in future directions.

3) The authors should include a simplified figure/cartoon (or graphical abstract) including the most important novel findings regarding the achievement gained with the technological advancements brought by their approach.

4) This reviewer personally misses some important topics in the introduction/discussion regarding the role of anthracyclines, among the most effective consequences, that the authors' findings can hold for humans and patients: Recently the two new proteins iRhoms 1 and 2 (inactive Rhomboid proteins, catalytically inactive members of the rhomboid family of intramembrane serine proteases) were identified as upstream ADAM-17 regulators, controlling the substrate selectivity of ADAM-17-dependent shedding. ADAMs are pivotal in orchestrating the junctional adhesion molecules dynamic and context-dependent expression with remarkable consequences in physiological and pathological hematopoiesis (please refer to PMID: 32354870).

5) The authors need to highlight what new information the review is providing to enhance the research in progress.

Author Response

Reviewer 3.

  1. The review is to summarize the work on this locus that has been performed over a time period of more than 20 years to give an overview of the evolutionary changes that have occurred in this locus over more than 400 million years and to identify the areas where the most interesting new finding may come. The data summarized here is coming from more than 20 original articles and where it is difficult for a person not directly familiar with field to have a clear picture of the major changes that has occurred in this locus over 400 million years and what that can tell us about the functional consequences of the large changes in gene numbers and specificities of these enzymes.
  2. Concerning the statement about no new data. I think this is valid only for original articles and not for Reviews. Reviews does in general only contain published material but in a new context, like hear, summarizing the work of more than 20 years and more than 20 articles to give an overview of the progress in the field and to be forward looking into future directions. If one adds new data then also a full description of Materials and Methods and also a Results section is needed to describe unpublished data in great detail. However, to mix old and new data in a review is normally not a common practice and makes it difficult with the inclusion of a lot of experimental detail which have a tendency to mask the message of the article and make it difficult for the reader.
  3. A very good suggestion with a graphical abstract and we have seriously discussed the possibility. The only thing which came up was an evolutionary tree where the different animal groups are represented and where the presence or absence of genes in the chymase locus are marked for each animal in the figure. However, after starting to make the drawing we realized that it would be too complicated for a reader to understand without a massive figure legend which removes the entire idea with a graphical abstract. So we could not come up with a good solution Sorry- but a good suggestion.
  4. Neither iRhoms nor ADAM-17 are encoded from the chymase locus and therefore outside of the scope of this review, which only involves genes of the chymase locus.
  5. In the end of the discussion we highlight the new experiments needed to enhance our understanding of the function and evolution of members of this locus and the difficulties we have had in analyzing the early amphibian and reptile members. However, as we also mention in the end of the discussion improvement in the technology may increase our chances to also obtain detailed information concerning these additional chymase locus members in a near future, as we have succeeded with the Catfish I and II which are highly specific proteases. The information concerning specificity of the reptile and amphibian enzymes would give us direct evidence for the first member of this locus.  Based on the alignment in figure 10 the first member appear to be granzyme B and that both the chymase and the dual chymase-tryptase (Cathepsin G) came in somewhat later by gene duplication and specificity changes. However, we need direct functional proof that this is the actual situation.

Round 2

Reviewer 1 Report

Dear Authors,

Thank you for correcting the manuscript according to my suggestions. The article now meets the requirements of the IJMS.

References to the literature in figure captions, to all manuscripts from which the original data and figures were obtained, will allow readers to get more context about how this data was obtained.

Great thanks for your detailed explanations regarding functional granzyme genes in chymase locus of the rat genome. Indeed, the increase in the sequence quality of the rat genome changes our understanding of granzyme genes in chymase locus.

I have no further suggestions for this manuscript. The manuscript can be approved for publication in IJMS.

Best regards

Reviewer 3 Report

The authors have clarified several of the questions I raised in my previous review. Unfortunately, most of the major problems have not been addressed by this revision.

For instance, simple rephrasing is not enough to encompass a large number of text similarities from other manuscripts from the authors. Moreover, novelty is not a simple characteristic of original research, but rather also characterize review if originally conducted (for instance, besides text similarities and revision, figures obtained or modified from the previously published manuscript, might be accepted, but in a bigger picture of the topic of interest, otherwise would just be a repetition).